# Dual-Rate Extended Kalman Filter Based Path-Following Motion Control for an Unmanned Ground Vehicle: Realistic Simulation

**DOI:** 10.3390/s21227557

**Published:** 2021-11-13

**Authors:** Rafael Carbonell, Ángel Cuenca, Vicente Casanova, Ricardo Pizá, Julián J. Salt Llobregat

**Affiliations:** Instituto Universitario de Automática e Informática Industrial, Universitat Politècnica de València, 46022 València, Spain; acuenca@isa.upv.es (Á.C.); vcasanov@isa.upv.es (V.C.); rpiza@isa.upv.es (R.P.); julian@isa.upv.es (J.J.S.L.)

**Keywords:** unmanned ground vehicle, Kalman filter, vehicle modeling and simulation

## Abstract

In this paper, a two-wheel drive unmanned ground vehicle (UGV) path-following motion control is proposed. The UGV is equipped with encoders to sense angular velocities and a beacon system which provides position and orientation data. Whereas velocities can be sampled at a fast rate, position and orientation can only be sensed at a slower rate. Designing a dynamic controller at this slower rate implies not reaching the desired control requirements, and hence, the UGV is not able to follow the predefined path. The use of dual-rate extended Kalman filtering techniques enables the estimation of the fast-rate non-available position and orientation measurements. As a result, a fast-rate dynamic controller can be designed, which is provided with the fast-rate estimates to generate the control signal. The fast-rate controller is able to achieve a satisfactory path following, outperforming the slow-rate counterpart. Additionally, the dual-rate extended Kalman filter (DREKF) is fit for dealing with non-linear dynamics of the vehicle and possible Gaussian-like modeling and measurement uncertainties. A Simscape Multibody™ (Matlab^®^/Simulink) model has been developed for a realistic simulation, considering the contact forces between the wheels and the ground, not included in the kinematic and dynamic UGV representation. Non-linear behavior of the motors and limited resolution of the encoders have also been included in the model for a more accurate simulation of the real vehicle. The simulation model has been experimentally validated from the real process. Simulation results reveal the benefits of the control solution.

## 1. Introduction

An unmanned ground vehicle (UGV) can be defined as a land-based vehicle that is capable of intelligent motion and action without human input [1]. UGVs can be used in a huge number of applications such as path tracking [2], storage [3], surveillance [4], transportation [5], in unstructured environments [6] (e.g., search and rescue operations [7], planetary exploration [8], agricultural works [9]), and so on. In our work, a path tracking application is developed, using a two-wheeled differential drive robot as UGV. In the path tracking problem, the controller is designed in order to ensure that the UGV is able to follow a predefined sequence of positions and orientations in the plane. The path tracking problem can be seen from two different points of view [10]: (i) path-following motion control, where the vehicle is required to converge to and follow the path without a temporal law; (ii) trajectory-tracking motion control, where the controller forces the vehicle to reach and follow a time parameterized reference (i.e., a geometric path with an associated timing law). In our particular case, no time constraints are required for the UGV, and hence, a path-following motion control will be carried out.

In the present approach, the celebrated Kalman filter is used in its extended version (see, e.g., [11]) with the aim of: (i) estimating the non-linear behavior of the UGV, providing not available (not measurable) variables, and reducing the possible process and measurement noise effect; (ii) fusing all the data provided by the different sensing devices (encoders and beacons) to be used in the control stage. Since every sensor may work at a different rate, which may be slower than the actuation (control) rate, a multi-rate extended Kalman filter may be needed. In our approach, rotational velocities are sensed by the encoders at the same rate as the control signal (fast rate), whereas position and orientation are sampled by the beacons at a rate *N*-times slower than the actuation one (slow rate). This leads to a dual-rate extended Kalman filter (DREKF). The proposed DREKF is featured by: (i) updating the filter gain at every time instant; (ii) resizing the dimension of the gain according to the number of available UGV outputs at every time instant. This number will depend on the different sensing rates.

Few works can be found on DREKF in literature. For instance, the DREKF is used to better estimate not available system states in biomedicine applications [12,13], in unmanned aerial vehicles [14], and for the simultaneous localization and map problem in robotics [15,16]. More related to UGV frameworks, in a recent work [17], a DREKF has been used to fuse output variables sensed at the slowest rate involved in the control system for a self-driving vehicle. In [17], different from the DREKF version stated in the present work, the Kalman filter gain is only updated at the slow rate, and hence, its dimension remains constant. To the best of the authors’ knowledge, the formulation of the DREKF proposed in the current work is novel in UGV frameworks.

In order to achieve a high reliability simulation, Simscape Multibody has been used. This multibody modeling tool is an extension of MATLAB/Simulink, where complex physical bodies and their interactions can be intuitively defined so as to realistically simulate system dynamics. Simscape Multibody has been employed in some recent studies. For instance, to perform further dynamics and control analysis in exoskeleton robots for gait rehabilitation [18], to prompt the design of the ankle joint for biped robots [19], for thrust evaluation in a quadrotor helicopter system in presence of wind fields [20], and more related to UGV frameworks, in [21] to simulate an anti-lock braking system -ABS- control for a vehicle, and in [22] to simulate different motion models for an omnidirectional mobile robot with Mecanum wheels. In our particular case, Simscape Multibody will be utilized to define some UGV dynamics such as contact forces, non-linear motor behavior, limited encoder resolution, etc, which are difficult to include in the kinematic and dynamic UGV representation. In this way, simulation results will accurately reproduce the expectable, real UGV behavior.

Summarizing, the main contributions of the work are:Consideration of DREKF, which enables one to:-Design a fast-rate dynamic controller capable of reaching the desired specifications for the UGV and precisely following the predefined path.-Generate fast-rate state estimates from slow-rate measurements to be supplied to the dynamic controller.-Face non-linear UGV dynamics and possible Gaussian-like modeling and measurement uncertainties.Development of a powerful simulation tool, which takes into account complex modeling aspects to realistically represent the UGV behavior.

The paper is organized as follows. Section 2 describes the problem scenario and the proposal of control solution. Section 3 is devoted to the formulation of the DREKF. Section 4 presents UGV modeling aspects and the simulation tool developed. Section 5 introduces the cases simulated and the results obtained, which are analyzed in detail via some cost indexes. Finally, some conclusions summarize the present work in Section 7.

## 2. Problem Scenario

The overall control scheme is depicted in Figure 1, where two main parts can be distinguished:Robot simulator, which contains some complex UGV modeling aspects as a result of using the features of the specialized Simscape Multibody simulation tool.Control structure, which includes a path tracking controller (in this case, the Pure Pursuit algorithm), an inverse kinematics computation block, a dynamic proportional integral (PI) controller, and a state estimator (in this case, the proposed DREKF).

The UGV is located together with different actuators, which introduce some non-linearity problems such as dead zone, friction, and saturationand sensors, which can introduce measurement noise and quantization. These are the aspects to be covered by the robot simulator to realistically model the UGV behavior.

The control structure is in charge of generating the path reference based on waypoints, and then, generating the consequent actions to control the actuators. The use of the DREKF enables one to cope with the non-linearities and noises that can appear in the UGV model. In addition, the DREKF is able to fuse output data sensed at two different rates in order to estimate the UGV state at the fastest rate.

The control structure uses two different periods: *T* as the estimation and control period, and the sensing period of the angular velocity of the wheel; and NT as the vehicle position and orientation sensing period, where $N\in N^{+}$ is the multiplicity between both periods. As commented, whereas the angular velocity output can be sensed by encoders at the fastest period *T*, the pose output will be more slowly sampled at period NT due to hardware constraints in the beacon system. Let us respectively denote ${\left(.\right)}_{k}^{T}$ and ${\left(.\right)}_{k}^{NT}$ as a *T*-period and an NT-period signal or variable, where k∈N are iterations at the corresponding period. In more detail, the control structure works as follows:At the current instant kT, the Pure Pursuit path tracking algorithm [23,24,25] generates velocity references ${\left(v^{ref},w^{ref}\right)}_{k}^{T}$ from a set of waypoints and the current pose estimation ${\left(\hat{X},\hat{Y},\hat{\psi }\right)}_{k}^{T}$. The set of waypoints is composed of reference positions $\left(X^{ref},Y^{ref}\right)$, a velocity constant reference $V^{ref}$, and a look ahead distance $L^{ref}$.The UGV inverse kinematics block transforms the velocity references ${\left(v^{ref},w^{ref}\right)}_{k}^{T}$ into dynamic references ${\left(w_{r}^{ref},w_{l}^{ref}\right)}_{k}^{T}$.From this dynamic reference ${\left(w_{r}^{ref},w_{l}^{ref}\right)}_{k}^{T}$ and the estimated angular velocities ${\left({\hat{w}}_{r},{\hat{w}}_{l}\right)}_{k}^{T}$, the dynamic PI controller computes the control signal to be applied to the UGV ${\left(u_{r},u_{l}\right)}_{k}^{T}$, which respectively are the control actions at period *T* for the right and left motors. These control actions will be applied under Zero Order Hold (ZOH) conditions.The UGV is equipped with a virtual beacon. Four fixed beacons are additionally placed on the walls of the simulation environment, emulating a beacon based indoor positioning system. The measurements ${\left(d_{1},d_{2},d_{3},d_{4}\right)}_{k}^{NT}$ are the distances between the virtual mobile beacon and the four fixed beacons, which are located in a known place with respect to the world system of the simulator. The simulation tool is able to work from these distances or, alternatively, from the direct pose information (X,Y,${\psi )}_{k}^{NT}$. Distances and pose can be equivalently deduced by applying the Pithagorean theorem (see, e.g., [26] and more details in Section 3).Any UGV output measurement ${\left(w_{r},w_{l}\right)}_{k}^{T}$, ${\left(d_{1},d_{2},d_{3},d_{4},\psi \right)}_{k}^{NT}$, or (X,Y,${\psi )}_{k}^{NT}$ may be disturbed by Gaussian noises, which will be created from a set of independent seeds in order to generate a reproducible pseudo-random noise. As a consequence, the experiments developed with the simulator will be reproducible under the same conditions.The system state estimate ${\left({\hat{w}}_{r},{\hat{w}}_{l},\hat{X},\hat{Y},\hat{\psi }\right)}_{k}^{T}$ is computed via the DREKF. The prediction step is generated at period *T* from the control actions ${\left(u_{r},u_{l}\right)}_{k}^{T}$. The correction step is also obtained at period *T*, but from data sensed at the two different periods, that is, ${\left(w_{r},w_{l}\right)}_{k}^{T}$, and ${\left(X,Y,\psi \right)}_{k}^{NT}$, or ${\left(d_{1},d_{2},d_{3},d_{4},\psi \right)}_{k}^{NT}$. More details can be found in Section 3.

## 3. Dual-Rate Extended Kalman Filter

As mentioned in previous sections, one of the main aims of the DREKF is the computation of fast-rate state estimates from slow-rate measurements, taking into consideration a non-linear representation of the UGV. So to do it, a linearization procedure is needed, which is based on the use of the Jacobian matrix (a matrix of partial derivatives). At every time step, this matrix is evaluated with current predicted states. Different from EKF and the DREKF presented in [17], the version of DREKF stated in this work resizes its Kalman filter gain in order to contemplate the different sensing rates. In other words, depending on the variables available at every sampling instant, the dimension of the output array, and consequently the dimension of the Kalman filter gain, are properly modified.

Next, in Section 3.1 the kinematic and dynamic UGV model considered by the DREKF for state estimation is exposed. The dynamic model will also be used to tune the PI controller. In Section 3.2, the DREKF algorithm will be formulated.

### 3.1. Kinematic and Dynamic UGV Modeling

The kinematic model represents the UGV velocity evolution in a fixed inertial frame. A discrete-time version of the model at period *T* can be deduced as follows [27]:(1)$$\begin{array}{cc} {\left(v_{r}\right)}_{k}^{T} & =r_{r}{\left(\omega_{r}\right)}_{k}^{T}\\ {\left(v_{l}\right)}_{k}^{T} & =r_{l}{\left(\omega_{l}\right)}_{k}^{T}\\ v_{k}^{T} & =\frac{{\left(v_{r}\right)}_{k}^{T}+{\left(v_{l}\right)}_{k}^{T}}{2}\\ \omega_{k}^{T} & =\frac{{\left(v_{r}\right)}_{k}^{T}-{\left(v_{l}\right)}_{k}^{T}}{2b}\\ X_{k}^{T} & =X_{k-1}^{T}+v_{k}^{T}T\cos\left(\psi_{k-1}^{T}+\omega_{k}^{T}T\right)\\ Y_{k}^{T} & =Y_{k-1}^{T}+v_{k}^{T}T\sin\left(\psi_{k-1}^{T}+\omega_{k}^{T}T\right)\\ \psi_{k}^{T} & =\psi_{k-1}^{T}+\omega_{k}^{T}T \end{array}$$

$\mathrm{for}\;k\in N_{\geq 1}$, where ${\left(v_{r}\right)}_{k}^{T},{\left(v_{l}\right)}_{k}^{T}$ are the linear velocities for each wheel, which are obtained from the corresponding rotational velocities ${\left(\omega_{r}\right)}_{k}^{T},{\left(\omega_{l}\right)}_{k}^{T}$, and radius $r_{r}$, $r_{l}$ of the wheels; $v_{k}^{T},\omega_{k}^{T}$ are respectively the UGV linear and rotational velocities, where *b* is half of the distance between the wheels; and ${\left(X,Y,\psi \right)}_{k}^{T}$ is the UGV position and orientation, being ${\left(X,Y,\psi \right)}_{0}^{T}$ the predefined, initial one.

The dynamic model represents the relation between the control signal for each motor $\left(u_{r},u_{l}\right)$ and the rotational velocity for each wheel. As a result of applying classical identification methods [28], this relation can be easily expressed as a continuous-time transfer function:(2)$$\begin{array}{c} G_{r}\left(s\right)=\frac{\omega_{r}\left(s\right)}{u_{r}\left(s\right)}\\ G_{l}\left(s\right)=\frac{\omega_{l}\left(s\right)}{u_{l}\left(s\right)} \end{array}$$

Using Z-transform at period *T*, the input–output plant model in (Equation 2) can be represented as a discrete-time transfer function:(3)$$\begin{array}{c} G_{r}^{T}\left(z\right)=\frac{\omega_{r}^{T}\left(z\right)}{u_{r}^{T}\left(z\right)}\\ G_{l}^{T}\left(z\right)=\frac{\omega_{l}^{T}\left(z\right)}{u_{l}^{T}\left(z\right)} \end{array}$$
being *z* the discrete *T* operator.

Alternatively, considering state-space representation, the model in (Equation 3) can take this form:(4)$$\begin{array}{c} \left\{\begin{array}{cc} {\left(x_{r}\right)}_{k+1}^{T} & =A_{r}{\left(x_{r}\right)}_{k}^{T}+B_{r}{\left(u_{r}\right)}_{k}^{T}\\ {\left(\omega_{r}\right)}_{k}^{T} & =C_{r}{\left(x_{r}\right)}_{k}^{T} \end{array}\right.\\ \left\{\begin{array}{cc} {\left(x_{l}\right)}_{k+1}^{T} & =A_{l}{\left(x_{l}\right)}_{k}^{T}+B_{l}{\left(u_{l}\right)}_{k}^{T}\\ {\left(\omega_{l}\right)}_{k}^{T} & =C_{l}{\left(x_{l}\right)}_{k}^{T} \end{array}\right. \end{array}$$
where ${\left(x_{r}\right)}_{k}^{T},{\left(x_{l}\right)}_{k}^{T}$ are respectively the process state for the motors of the right and left wheels, and $A_{r},B_{r},C_{r}$ and $A_{l},B_{l},C_{l}$ are matrices with suitable dimensions.

Following classical control techniques [29], the dynamic, PI controller can be tuned from any of the models in (Equation 2), (Equation 3), or (Equation 4). A more detailed dynamic UGV modeling can be found in [27].

### 3.2. DREKF Algorithm

From (Equation 1) and (Equation 4), the global model for the UGV can be obtained. This model will be used by the DREKF to estimate the non-available UGV position and orientation. In order to do it, the next state-space representation may be considered:(5)$$\begin{array}{c} \left\{\begin{array}{c} \xi_{k}^{T}=f\left(\xi_{k-1}^{T},{\left(n_{1}\right)}_{k-1}^{T},u_{k-1}^{T}\right)\\ z_{k}^{T}=h\left(\xi_{k}^{T},{\left(n_{2}\right)}_{k}^{T}\right) \end{array}\right. \end{array}$$
where $\xi_{k}^{T}$ is the UGV state, which is composed of ${\left[{\left(\omega_{r},\omega_{l},X,Y,\psi \right)}_{k}^{T}\right]}^{⊤}$, denoting ${\left[\cdot \right]}^{⊤}$ as the transpose function; the control signal is $u_{k-1}^{T}={\left[{\left(u_{r},u_{l}\right)}_{k-1}^{T}\right]}^{⊤}$; and ${\left(n_{1}\right)}_{k-1}^{T}$ and ${\left(n_{2}\right)}_{k}^{T}$ are respectively possible process and measurement noises, which are both assumed to be zero mean multivariate Gaussian noises with covariance $Q_{k}^{T}$ and $R_{k}^{T}$, respectively. Regarding the output $z_{k}^{T}$, it consists of $z_{k}^{T}={\left[{\left(\omega_{r},\omega_{l},X,Y,\psi \right)}_{k}^{T}\right]}^{⊤}$ at the sampling instants coinciding with the slower rate (k=NT), and $z_{k}^{T}={\left[{\left(w_{r},w_{l}\right)}_{k}^{T}\right]}^{⊤}$ at the rest of the fast-rate sampling instants (k≠NT).

Let us notate ${\hat{\xi }}_{j|i}^{T}$ as the state estimated for the instant jT at the instant iT. Then, the prediction and correction steps of the DREKF are defined as follows:Prediction of the next state ${\hat{\xi }}_{k|k-1}^{T}$ and propagation of the covariance $P_{k|k-1}^{T}$:
(6)$$\begin{array}{cc} {\hat{\xi }}_{k|k-1}^{T} & =f\left({\hat{\xi }}_{k-1|k-1}^{T},{\left(n_{1}\right)}_{k-1}^{T},u_{k-1}^{T}\right)\\ P_{k|k-1}^{T} & =A_{k}^{T}P_{k-1|k-1}^{T}{\left[A_{k}^{T}\right]}^{⊤}+L_{k}^{T}Q_{k-1}^{T}{\left[L_{k}^{T}\right]}^{⊤} \end{array}$$$\mathrm{for}\;k\in N_{\geq 1}$, where ${\hat{\xi }}_{0}^{T}=E\left[\xi_{0}^{T}\right]$, E[·] being the expectation, and $P_{0}^{T}=E[\left(\xi_{0}^{T}-E\left[\xi_{0}^{T}\right]\right)$${\left[\left(\xi_{0}^{T}-E\left[\xi_{0}^{T}\right]\right)\right]}^{⊤}]$, and where $A_{k}^{T}$ and $L_{k}^{T}$ are Jacobian matrices computed in order to respectively linearize the process model about the current state and about the process noise:
(7)$$\begin{array}{cc} A_{k}^{T} & ={\left.\frac{\partial f}{\partial \xi }\right|}_{{\hat{\xi }}_{k-1|k-1}^{T},{\left(n_{1}\right)}_{k-1}^{T},u_{k-1}^{T}}\\ L_{k}^{T} & ={\left.\frac{\partial f}{\partial n_{1}}\right|}_{{\hat{\xi }}_{k-1|k-1}^{T},{\left(n_{1}\right)}_{k-1}^{T},u_{k-1}^{T}} \end{array}$$Prediction of the future output ${\hat{z}}_{k}^{T}$, being ${\hat{z}}_{k}^{T}={\left[{\left({\hat{\omega }}_{r},{\hat{\omega }}_{l}\right)}_{k}^{T}\right]}^{⊤}$ for k≠NT, and ${\hat{z}}_{k}^{T}={\left[{\left({\hat{\omega }}_{r},{\hat{\omega }}_{l},\hat{X},\hat{Y},\hat{\psi }\right)}_{k}^{T}\right]}^{⊤}$ for k=NT:
(8)$${\hat{z}}_{k}^{T}=h\left({\hat{\xi }}_{k|k-1}^{T},{\left(n_{2}\right)}_{k}^{T}\right)$$Computation of the Kalman filter gain $K_{k}^{T}$:
(9)$$K_{k}^{T}=P_{k|k-1}^{T}{\left[H_{k}^{T}\right]}^{⊤}{\left(H_{k}^{T}P_{k|k-1}^{T}{\left[H_{k}^{T}\right]}^{⊤}+M_{k}^{T}R_{k}^{T}{\left[M_{k}^{T}\right]}^{⊤}\right)}^{-1}$$
where, depending on the time instant *k*, every matrix takes suitable dimensions to fit each option for the output array ${\hat{z}}_{k}^{T}$. $H_{k}^{T}$ and $M_{k}^{T}$ are the Jacobian matrices calculated in order to respectively linearize the output model about the predicted next state and about the measurement noise:
(10)$$\begin{array}{cc} H_{k}^{T} & ={\left.\frac{\partial h}{\partial \xi }\right|}_{{\hat{\xi }}_{k|k-1}^{T},{\left(n_{2}\right)}_{k}^{T}}\\ M_{k}^{T} & ={\left.\frac{\partial h}{\partial n_{2}}\right|}_{{\hat{\xi }}_{k|k-1}^{T},{\left(n_{2}\right)}_{k}^{T}} \end{array}$$Correction of the state ${\hat{\xi }}_{k|k}^{T}$ and correction of the covariance $P_{k|k}^{T}$:
(11)$$\begin{array}{cc} {\hat{\xi }}_{k|k}^{T} & ={\hat{\xi }}_{k|k-1}^{T}+K_{k}^{T}\left(z_{k}^{T}-{\hat{z}}_{k}^{T}\right)\\ P_{k|k}^{T} & =K_{k}^{T}R_{k}^{T}{\left[K_{k}^{T}\right]}^{⊤}+\left(I-K_{k}^{T}H_{k}^{T}\right)P_{k|k-1}^{T}{\left[(I-K_{k}^{T}H_{k}^{T})\right]}^{⊤} \end{array}$$

### 3.3. Beacon Measurement Model

Instead of using direct measurements of position ${\left(X,Y\right)}_{k}^{T}$ at the instants k=NT, distances ${\left(d_{1},d_{2},d_{3},d_{4}\right)}_{k}^{T}$ from the beacon system may be considered. Then, with the aim of obtaining (Equation 8) for the instants at k=NT, a straightforward conversion based on the Pythagorean theorem is required so as to compute the future distances ${\left[{\left({\hat{d}}_{1},{\hat{d}}_{2},{\hat{d}}_{3},{\hat{d}}_{4}\right)}_{k}^{T}\right]}^{⊤}$ from the predicted position ${\left[{\left(\hat{X},\hat{Y}\right)}_{k|k-1}^{T}\right]}^{⊤}$. The position of the beacons on the base reference frame $(\left(X_{1},Y_{1}\right),\left(X_{2},Y_{2}\right),$$\left(X_{3},Y_{3}\right),\left(X_{4},Y_{4}\right))$ are known, and the height $Z_{fb}$ with respect to the ground plane is the same for all the fixed beacons. The virtual mobile beacon is parallel with respect to the ground plane $Z_{mb}$ and, consequently, the fixed beacons and the mobile beacon are parallel too. So, the distance between them is constant $Z_{d}={\left(Z_{fb}-Z_{mb}\right)}^{2}$. Finally, the conversion yields [26]:(12)$${\left[\begin{array}{c} {\hat{d}}_{1}\\ {\hat{d}}_{2}\\ {\hat{d}}_{3}\\ {\hat{d}}_{4} \end{array}\right]}_{k}^{T}=\left[\begin{array}{c} \sqrt{{\left(X_{1}-{\hat{X}}_{k|k-1}^{T}\right)}^{2}+{\left(Y_{1}-{\hat{Y}}_{k|k-1}^{T}\right)}^{2}+Z_{d}}\\ \sqrt{{\left(X_{2}-{\hat{X}}_{k|k-1}^{T}\right)}^{2}+{\left(Y_{2}-{\hat{Y}}_{k|k-1}^{T}\right)}^{2}+Z_{d}}\\ \sqrt{{\left(X_{3}-{\hat{X}}_{k|k-1}^{T}\right)}^{2}+{\left(Y_{3}-{\hat{Y}}_{k|k-1}^{T}\right)}^{2}+Z_{d}}\\ \sqrt{{\left(X_{4}-{\hat{X}}_{k|k-1}^{T}\right)}^{2}+{\left(Y_{4}-{\hat{Y}}_{k|k-1}^{T}\right)}^{2}+Z_{d}} \end{array}\right]$$

## 4. UGV Modeling. Simulation Tool

### 4.1. Preliminary Considerations

The proposed path-following algorithm must be evaluated to prove its validity and how the desired trajectory is followed with more precision than when using conventional odometry based strategies. The best way to do this is by using a real vehicle, following a real trajectory over a real floor. The goodness of the proposed solution must show that the comparison between the desired trajectory and the real one is improved. The main problem of using a real vehicle is that it is not easy to measure the real position and orientation (X,Y,ψ). Several approaches can be used to do so:Vision-based system: Using a zenithal camera the position and orientation of the vehicle can be measured. The camera sees what the vehicle is doing from above and can provide information about the time evolution of the (X,Y) coordinates and angular position ψ. The main drawback is that the vehicle must lie under the camera, which is a great limitation for the desired trajectory.GPS: This is probably the best way to measure the real trajectory to be compared with the desired one. The main drawback is that it must be used outdoors, and the resolution is not suitable to be used with small vehicles as the one is being used in this work.Beacons: Suitable to be used indoors it will probably be the best solution. As mentioned in previous sections, position (X,Y) can be measured based on the distance $d_{i},\left(i=1\ldots n\right)$ from the vehicle to several (*n*) fixed beacons. Orientation ψ can be measured based on information provided by an inertial measurement unit. The drawback is the measurement noise and the lack of precision when using small vehicles.

When it is not possible to use the real system, simulation tools can give a valuable helping hand. So, an alternative solution to the real vehicle is to use a simulation model to get the ’real’ position and orientation of the vehicle. The easiest way is to use conventional transfer functions to simulate the behavior of the motors that move the wheels and conventional odometry to estimate the trajectory. With this approach it is assumed that ideal motors and ideal wheels are used, which is far away from the real world vehicle. Then, a more realistic simulation must be utilized. Non-linearities involved in the motor behavior and not ideal friction between the wheels and the ground must be included in the simulation model, as they have a great influence on the position and orientation (X,Y,ψ) of the real vehicle.

From this kind of simulation model, it is easy to give an answer to ‘what-if’ questions. Different control strategies can be evaluated, and it can be studied how electrical and/or mechanical disturbances affect the path-following behavior. For example, it can be shown how the behavior is modified if one of the wheels changes its friction coefficient with the ground or one of the wheels is slightly more worn out than the other.

Matlab/Simulink Simscape Library has been used in this work to achieve this goal. Simscape Multibody has been utilized to simulate the rotational movement of the wheels and the forces and torques caused by the friction of the wheels against the ground.

### 4.2. Modeling Aspects. Tool Description

The main goal of this section is to reach an accurate simulation model for the UGV, which is the Lego^®^ Mindstorms^®^ EV3. This model has been developed in two stages. The first one involves modeling the Lego DC motor, used in the real vehicle without considering the friction of the wheels with the ground (Section 4.2.1). Simscape Electrical library could have been used to build this model. However, as the behavior of the motor is quite linear, it can be fairly well modeled with some conventional Simulink blocks to include some minor non-linearities (saturation, dead zone, quantization). The output of this electrical part is the torque applied by the motor to the wheel. This torque has been applied to a Simscape Multibody revolute joint to generate the angular velocity of the wheel. This simulated velocity can be compared with the real one, measured by the encoder in the real vehicle. In the second stage, mechanical aspects like forces and torques generated by the wheels against the ground have been modeled using Simscape Multibody library (Section 4.2.2). By means of Simulink Sensitivity Analysis Tool, the value of every parameter included in the simulation model has been tuned (Section 4.3).

#### 4.2.1. Wheels-on-the-Air Simulation Model

In this section the Lego DC motor that uses the real vehicle has been modeled to get an accurate simulation of the relationship between the applied input and the angular velocity of each motor. Simscape Electrical library can be used to model electromechanical systems as a DC motor. Nevertheless, as the Lego DC motor has a strong linear behavior it can be modeled in a simpler way by using conventional Simulink blocks.

Figure 2 shows the wheels-on-the-air simulation model for the UGV. The Lego DC motor has a unitless control action input within the range [−100,100] and the output provided is the angular velocity in $\frac{rad}{s}$. The input has been transformed into torque constant $K_{\tau }$. The torque is applied to a revolute joint that makes the wheel turn with friction constant *B*. The wheel is modeled by its computer aided design (CAD) file which provides information of its geometry, inertia *J*, and density. Both the torque and the friction constant enable the inclusion of the non-linear behavior of the motor. Dead zone, saturation, and a unitary quantification input have also been included in the model. The parameters of this model have been experimentally determined from the real response of the Lego DC motor following the procedure in Section 4.3 (see Table 1).

Figure 3a compares the input–output characteristic response of the real Lego DC motor with the proposed simulation model, where the practically linear behavior of the motor is observed. Figure 3b compares the open-loop response of the real motor with the proposed model, showing an accurate motor modeling.

Then, the DC motor model developed via Simscape Multibody can be used to precisely simulate the real motor behavior. However, for controller design purposes, a linear transfer function for the DC motor such as in (Equation 2) is available, which enables one to simply design the PI controller. In our particular case, and after applying classical identification techniques [28] from real process experiments, (Equation 2) leads to $G_{r}\left(s\right)=G_{l}\left(s\right)=\frac{K}{\tau s+1}$, where K=0.1481 and τ=0.064. Then, following classical control design procedures [29], the controller’s parameters Kp and Ti can be adjusted to get a closed-loop response with the shortest settling time, no overshoot, and no position error, yielding Kp=0.72 and Ti=0.064. Figure 4 shows the velocity control implemented for a sampling time T=0.1s.

To get a better approximation to the real system, the resolution of the real encoder has been included in the velocity feedback. The encoder provides 360 counts per revolution, which means that the minimal angle that can be measured is $\frac{2\pi }{360}=0.0175\;rad$. The minimal increment of velocity is $\frac{2\pi }{360T}=0.175\;\frac{rad}{s}$. Figure 5 shows the closed-loop response from the simulation model. In the detail, the effect of the encoder resolution can be observed.

#### 4.2.2. Wheels-on-the-Ground Simulation Model

The simulation model from the first stage (previous section) provides information about the angular movement of the wheels. However, the vehicle is useless if the wheels do not contact the ground. In this second stage, forces and torques caused by the friction of the wheels against the ground are included to move the simulated vehicle. Contact forces between the wheel and the floor will generate forces and torques that make the vehicle move and turn, describing a certain trajectory. This is an improved version of the classical odometry algorithms closer to the real behavior.

The simulation model will take into account the inertia parameters (masses and moments of inertia) which depend on the density and geometry of the different parts of the vehicle. It will also consider the friction parameters (static and dynamic friction) which depend on the materials and shapes of the contact surfaces. Simscape Multibody library has been used to build this mechanical part of the simulation model, which can be seen in Figure 6 and Figure 7.

Figure 6 shows the complete UGV model composed of a body, a support ball, a mobile beacon, and two wheels with a 6 degree-of-freedom joint, allowing a free 3D space movement. This model includes a transform sensor for measuring the position and orientation of the robot with respect to the environment’s world system. Figure 7 shows the full environment system composed of walls, floor, contact of the UGV with the walls and with the floor, world frame with gravity, and beacon subsystem.

Starting with the CAD model of the different parts of the vehicle, they are assembled in Simscape Multibody to build the complete vehicle (depicted in Figure 8a). Assigning the appropriated weight to the parts, the application can calculate the density, position of the center of gravity, and moments of inertia. With this information, the torque generated by the DC motor is converted into an angular movement of the wheel. As the wheels turn, they graze the ground and this friction causes forces and torques that make the vehicle move and turn, as it happens in the real world. Additionally, the robot includes a steel support ball that produces another friction point when contacting with the ground, yielding the consequent dynamic and static friction coefficients.

The Simscape simulation model provides measures of the *X* and *Y* coordinates and the orientation angle ψ. This information represents ideal, perfect positioning data. It may only be used for comparison aims. For real control purposes, measurement noise is usually added in order to emulate the information that would be provided by a GPS or any other location device in the real vehicle. From the revolute joints that connect the wheels with the chassis, angular velocity of the wheels $\left(w_{r},w_{l}\right)$ can be measured as it can be by using encoders in the real vehicle. Indoor positioning based on ultrasonic and radio beacon system is simulated. The beacon system provides distances $\left(d_{1},d_{2},d_{3},d_{4}\right)$ between fixed beacons mounted on the walls with known position and mobile beacon equipped on the robot. Figure 8b shows the simulation environment with measurements, robot, walls, beacon system, and trajectory plotted on the floor.

### 4.3. Parameter Adjustments

In this section, the parameters used in the simulator to emulate the real UGV behavior will be tuned. These parameters are velocity-torque constant, revolute joint friction coefficient, and friction coefficients between wheels and ground and between steel ball and ground. The method applied to adjust the parameters is based on the concept of Data-Driven Modeling [30].

From a series of experiments of the real process with input and output data, the concept of Data-Driven Modeling is applied for fitting the simulator parameters by minimizing the output error, that is, the error between real output data and simulated output data for the same input reference. In the experiments carried out, the reference input is a constant angular velocity ${\left(w_{r}^{ref},w_{l}^{ref}\right)}_{k}^{T}$ injected to the PI controller, and the outputs are the angular velocities ${\left(w_{r},w_{l}\right)}_{k}^{T}$, position ${\left(X,Y\right)}_{k}^{T}$, and Euclidean distance with respect to the starting point, say, $d_{k}^{T}$. The cost function used to minimize is the average of the output error. The main aim is to be able to emulate noise and non-linearities from the real process data by means of the non-linear modeling aspects included in the simulator such as static and dynamic friction with ground, limited encoder resolution, etc.

Simulink Sensitivity Analysis Tool is used to fit the parameters. For each parameter, a searching space is generated by defining a range and a distribution function (see Figure 9a). For example, the friction parameters are defined to follow a uniform random distribution, which ranges from 0 to 1.2 according to the materials and contact elements (rubber with ground or steel with ground). For each set of parameters, the tool performs a simulation.

Figure 9b shows the results obtained for a straight-line experiment. For each set of parameters, the output error is presented. Its minimum value is given by the *y*-axis lowest value depicted in each plot.

From these results, to improve the parameter adjustment, a second experiment is carried out considering a circular trajectory. After adjusting the simulator parameters from the results obtained in both experiments, a comparison between simulation and reality is depicted in Figure 10a,b. As shown for the circular trajectory, the simulated outputs satisfactorily emulate the real ones.

The motor parameters identified for the UGV model simulator by means of Simulink Sensitivity Analysis Tool were presented in Table 1. The friction coefficients identified are shown in Table 2.

Other important parameters used in the simulation are:-The geometric parameters of the UGV in (Equation 1), that is, the wheel radius $r_{r}=r_{l}=0.028$ m and the half track width between wheels b=0.068 m.-The parameters used for the pure pursuit algorithm: velocity constant reference $V^{ref}=0.1$ m/s, and look ahead distance $L^{ref}=0.2$ m.-The Gaussian noises are generated by considering zero mean μ=0 and variance $\sigma^{2}=10^{-4}$.-The solver chosen is Ode4 Runge–Kutta, running at a fixed step of 0.1 ms.

## 5. Simulation

This section is organized as follows. The cases simulated are defined in Section 5.1. Some cost indexes are formulated in Section 5.2 with the aim of being used to better assess the results obtained in Section 5.2.

### 5.1. Cases Evaluated

These are the cases studied in the simulation tool:Direct pose and angular velocity: in this experiment, output measurements $(w_{r},w_{l},$${X,Y,\psi )}_{k}^{T}$ are directly sampled from the simulator block at different periods *T* = 0.1 s, *T* = 0.2 s, and *T* = 0.5 s. No noise is considered.Odometry: in this case, only angular velocities ${\left(w_{r},w_{l}\right)}_{k}^{T}$ are sampled at *T* = 0.1 s. Pose data ${\left(X,Y,\psi \right)}_{k}^{T}$ are estimated by odometry. Ideally, no noise may be considered. However, in real environments, where for instance an encoder may be needed to take the velocities, some measurement noise may appear. In this simulation, two cases are considered: without noise and with noise (assuming additive Gaussian noises and limited encoder resolution).Dual-rate Extended Kalman Filter: this is the case exposed in Section 2. Gaussian noises are assumed in pose ${\left(X,Y,\psi \right)}_{k}^{NT}$ and velocities ${\left(w_{r},w_{l}\right)}_{k}^{T}$. Two options are simulated: N=10 and N=50.Dual-rate Extended Kalman Filter with beacon distances: this is the case stated in Section 3.2. Gaussian noises are assumed in pose $(d_{1},d_{2},d_{3},$$d_{4}{)}_{k}^{NT}$ and velocities ${\left(w_{r},w_{l}\right)}_{k}^{T}$. For the sake of clarity, only the option for N=10 will be presented.

The reference trajectories to be tracked are a square and the Lissajous curve shown in Figure 11.

### 5.2. Cost Indexes for Performance Assessment

To better quantify the results to be shown in the Section 5.2, three cost indexes will be used. These indexes evaluate control performance for every case simulated with the aim of making the comparison easier. The cost indexes are:$J_{1}$, which is based on the $\ell_{2}$-norm, and its goal is to provide a measure (in meters) about how accurately the path is followed:
(13)$$J_{1}=\frac{1}{l}\sum_{k=1}^{l}\min_{1\leq k^{\prime }\leq l}\sqrt{{\left(X_{k}^{T}-{\left(X_{ref}\right)}_{k^{\prime }}^{T}\right)}^{2}+{\left(Y_{k}^{T}-{\left(Y_{ref}\right)}_{k^{\prime }}^{T}\right)}^{2}}$$
where *l* is the number of iterations at period *T* required by the UGV to reach the final point of the path, ${\left(X,Y\right)}_{k}^{T}$ is the current UGV position, and ${\left(X_{ref},Y_{ref}\right)}_{k^{\prime }}^{T}$ is the nearest kinematic position reference to the current UGV position. It is worth noting that, despite using a dual-rate control scheme, the position data may be available at period *T* (intersample behavior; see, e.g., [31]) in the simulator environment.$J_{2}$, which is based on the $\ell_{\infty }$-norm and is defined to know the maximum difference (in meters) between the desired path and the current UGV position:
(14)$$J_{2}=\max_{1\leq k\leq l}\left\{\min_{1\leq k^{\prime }\leq l}\sqrt{{\left(X_{k}^{T}-{\left(X_{ref}\right)}_{k^{\prime }}^{T}\right)}^{2}+{\left(Y_{k}^{T}-{\left(Y_{ref}\right)}_{k^{\prime }}^{T}\right)}^{2}}\right\}$$$J_{3}$, which measures the total amount of time (in seconds) elapsed to arrive at the final destination:
(15)$$J_{3}=lT$$

## 6. Results

Figure 12 and Figure 13 show the results obtained for the square and Lissajous curve references, respectively. Table 3 presents the cost indexes calculated for every case, where the value *∞* means the UGV is not able to track the desired trajectory.

The comparison takes the direct pose case at T=0.1 s (*D.P. 0.1*) as the desired, nominal performance, because it reaches the best (lowest) cost index values (as expected). Then:The direct pose case at T=0.2 s (*D.P. 0.2*) worsens its behavior with respect to the nominal case, since the trajectory presents oscillations, which is confirmed by the clear increase of every cost index. On average, $J_{1}$, $J_{2}$, and $J_{3}$ increase their values 333%, 118%, and 45%, respectively.The direct pose case at T=0.5 s (*D.P. 0.5*) presents an unstable response, not being able to track the path in any case.The odometry case without noise at T=0.1 s (*Odom*) does not show so many oscillations like the *D.P. 0.2* case, but the tracking seems to be not so accurate as in the nominal (*D.P. 0.1*) case. This analysis is corroborated by the cost indexes, which show a worsening with respect to the nominal case: $J_{1}$, $J_{2}$, and $J_{3}$ are respectively increased 90%, 59%, and 2%, which are lower increases than in the *D.P 0.2* case. The worsening is due to the addition of a systematic numerical error over time that typically appears when odometry is used.The odometry case with noise at T=0.1 s (*Odom N*) depicts a considerable path tracking worsening for the square reference, and is incapable of following the Lissajous curve. The cost indexes confirm this statement, since they are highly increased or *∞*, respectively.The DREKF case with N=10 (*DREKF 10*) shows an accurate path tracking, despite having scarce pose measurements (10 times less), and it assumes noise and non-linearities. The cost indexes indicate the achievement of satisfactory control properties, since $J_{1}$, $J_{2}$, and $J_{3}$ are slightly worsened with respect to the nominal case (32%, 23%, and 12%, respectively), and even $J_{1}$ and $J_{2}$ outperform the *Odom* case.The DREKF case with N=50 (*DREKF 50*) presents a worse response than *DREKF 10*, as expected (*DREKF 50* is provided with 5 times less measurements). The cost indexes $J_{1}$ and $J_{2}$ with respect to the *DREKF 10* case are increased 207% and 179%, respectively; $J_{3}$ is very similar. Despite having five times fewer measurements, if *DREKF 50* is compared with *D.P. 0.2*, both cases reach similar cost indexes. The main difference between them obeys the way of path tracking: whereas the *D.P. 0.2* case presents oscillations, the *DREKF 50* case depicts a smooth trajectory with underdamped response.The DREKF case with N=10 using beacon distances (*DREKF-D*) depicts a quite close response to the *DREKF 10* case, which is confirmed by achieving very similar cost index values.

As a summary, the proposed DREKF strategy enables one to reach a satisfactory control performance, similar to the nominal one for a lower *N*, despite managing noisy and scarce data and existing process non-linearities.

To check the possibilities and power of the simulation tool developed, the next link to one video that shows the cases evaluated is provided: https://1drv.ms/v/s!AgyvxPGH2rA4dMtTjtNeWQwmswo?e=13zXaP, accessed on 11 September 2021.

## 7. Conclusions

The proposed Dual-Rate Extended Kalman Filter solves the problem of estimating the state of a UGV from output measurements sensed at different periods and is able to reach a satisfactory path tracking, despite having scarce UGV output data and existing non-linearities and noises. The data-driven modeling via Simulink Sensitivity Analysis Tool is a valid option for fitting the parameters of the simulator from real data experiments. Model-based optimization techniques and the simulator developed with Simscape Multibody are combined to provide a useful method for tuning key parameters. The simulator developed results in a powerful tool for making new realistic experiments in future works, where any other kind of UGV and control structure may be easily integrated. 

## Figures and Tables

**Figure 1 sensors-21-07557-f001:**
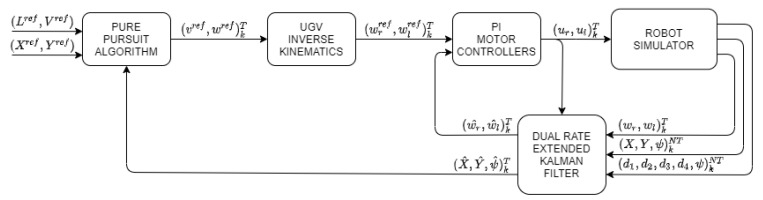
Control structure with DREKF estimator for the UGV.

**Figure 2 sensors-21-07557-f002:**
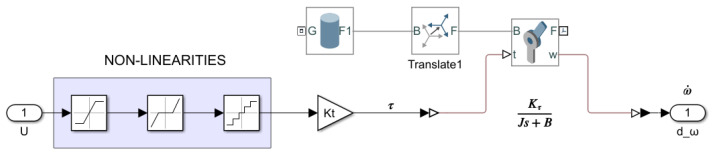
Simscape wheels-on-the-air simulation model.

**Figure 3 sensors-21-07557-f003:**
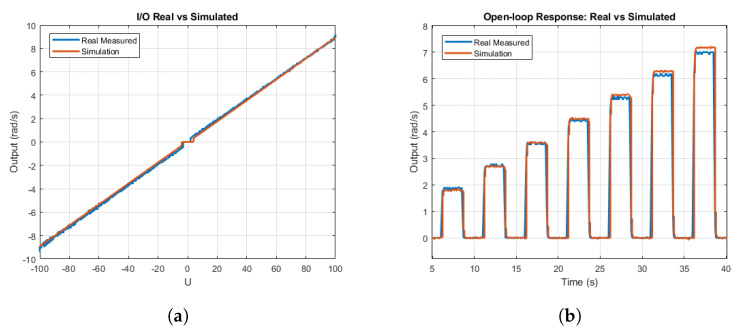
DC motor response: real vs. simulated. (**a**) Input–output characteristic response. (**b**) Open–loop step response.

**Figure 4 sensors-21-07557-f004:**
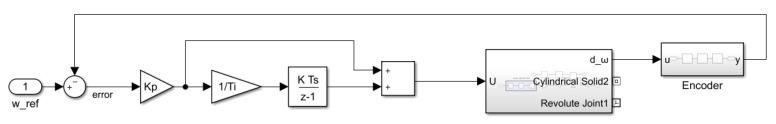
PI controller with motor model.

**Figure 5 sensors-21-07557-f005:**
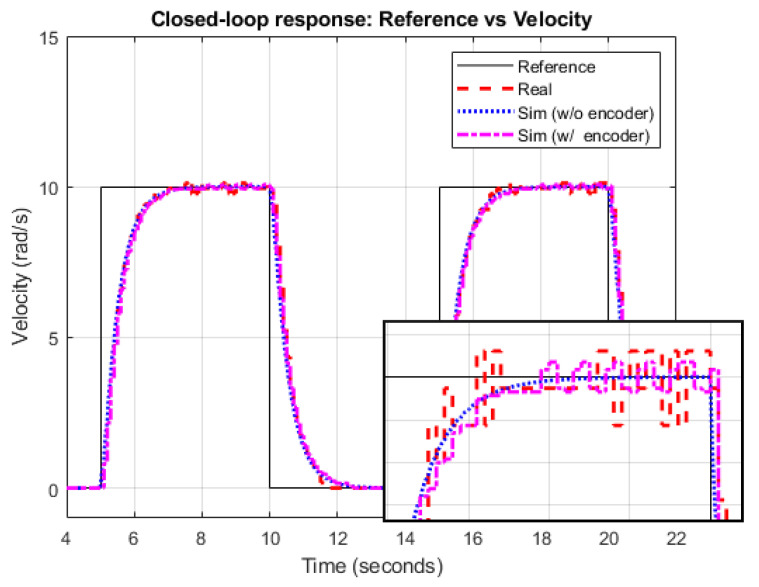
Closed-loop response.

**Figure 6 sensors-21-07557-f006:**
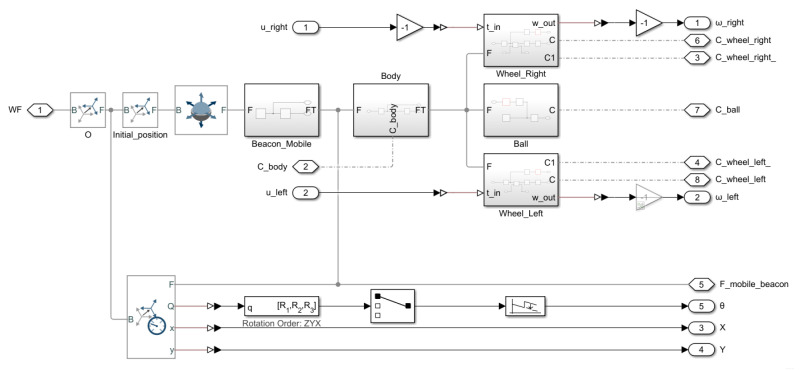
Simscape UGV model.

**Figure 7 sensors-21-07557-f007:**
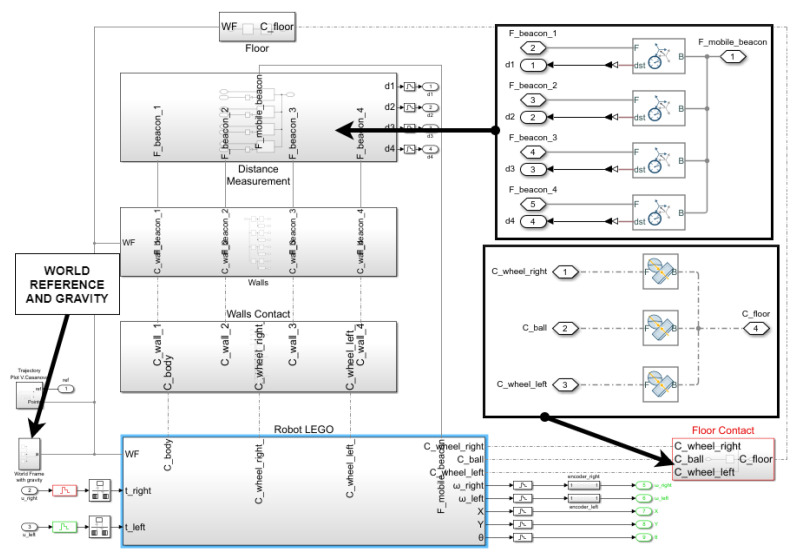
Simscape wheels-on-the-ground simulation model and environment.

**Figure 8 sensors-21-07557-f008:**
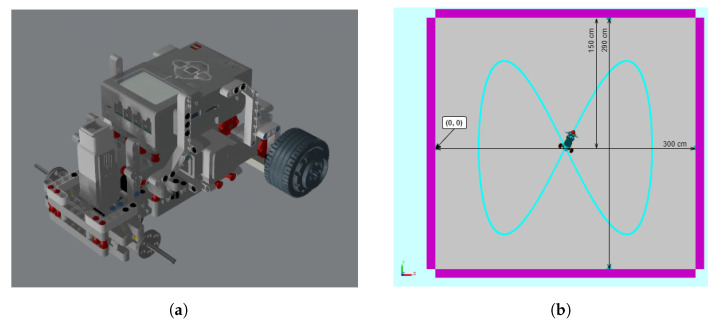
Model and environment. (**a**) CAD UGV model. (**b**) Simscape simulation environment.

**Figure 9 sensors-21-07557-f009:**
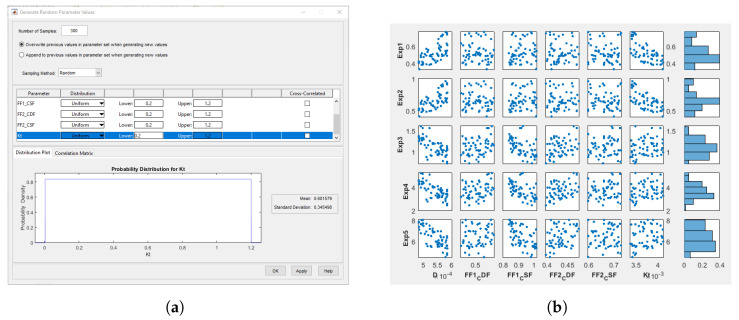
Sensitivity Analysis Tool. (**a**) Uniform random distribution generation. (**b**) Results of the experiment.

**Figure 10 sensors-21-07557-f010:**
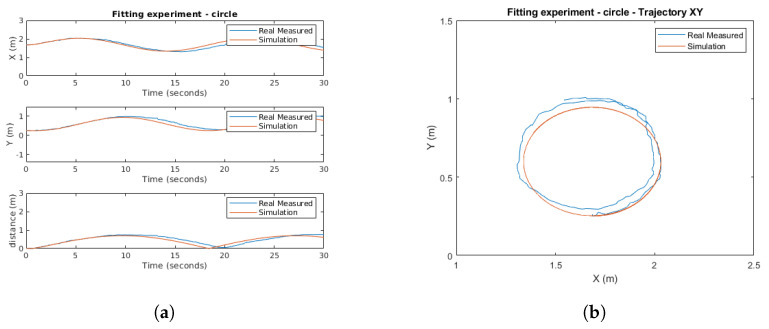
Circular trajectory. (**a**) Output with respect to time. (**b**) X-Y plane.

**Figure 11 sensors-21-07557-f011:**
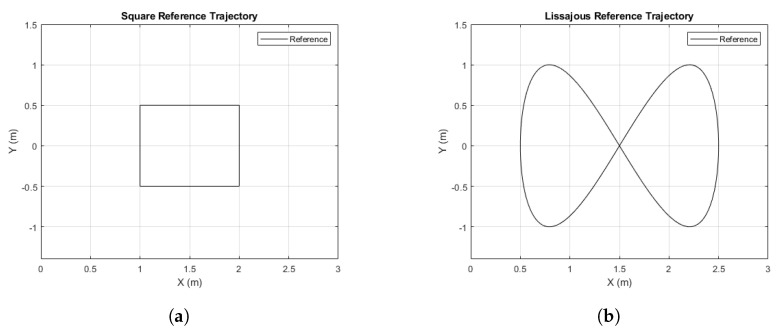
Reference trajectories. (**a**) Square reference. (**b**) Lissajous curve reference.

**Figure 12 sensors-21-07557-f012:**
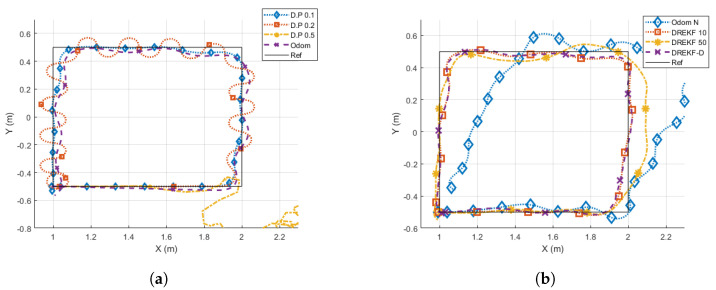
Results for a square reference. (**a**) *D.P* and *Odom* options. (**b**) *Odom N* and *DREKF* options.

**Figure 13 sensors-21-07557-f013:**
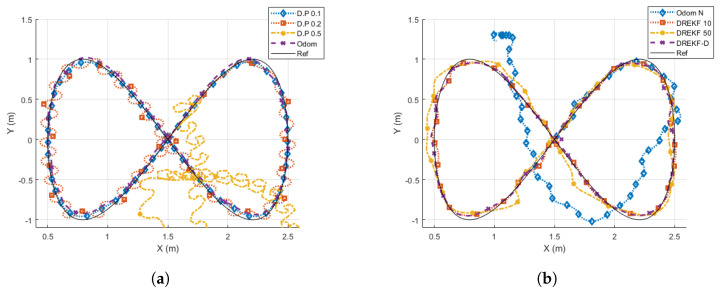
Results for a Lissajous reference. (**a**) *D.P* and *Odom* options. (**b**) *Odom N* and *DREKF* options.

**Table 1 sensors-21-07557-t001:** Motor parameters.

Parameter	Value
Torque constant ($K_{\tau }$)	0.00374 Nm
Damping coefficient (*B*)	5.3473 $\times 10^{-4}$ $\frac{\mathrm{Nms}}{deg}$
Moment of inertia (*J*)	1.38083 $\times 10^{-5}$ kgm${}^{2}$

**Table 2 sensors-21-07557-t002:** Friction coefficients.

Parameter	Value
Static friction coeff. rubber-ground	0.9285
Dynamic friction coeff. rubber-ground	0.5059
Static friction coeff. steel-ground	0.6619
Dynamic friction coeff. steel-ground	0.4349

**Table 3 sensors-21-07557-t003:** Cost index results.

Reference	Square			Lissajous		
Experiment	$J_{1}$	$J_{2}$	$J_{3}$	$J_{1}$	$J_{2}$	$J_{3}$
Direct pose T=0.1s (*D.P. 0.1*)	0.01251	0.04947	40.2	0.01259	0.04513	92.9
Direct pose T=0.2s (*D.P. 0.2*)	0.03417	0.09590	57.0	0.03688	0.10901	138.2
Direct pose T=0.5s (*D.P. 0.5*)	*∞*	*∞*	*∞*	*∞*	*∞*	*∞*
Odometry w/o noise (*Odom*)	0.02695	0.07053	41.1	0.02077	0.07999	94.3
Odometry w noise (*Odom N*)	0.10844	0.36137	43.7	*∞*	*∞*	*∞*
DREKF N=10 (*DREKF 10*)	0.01759	0.06215	45.1	0.01551	0.05451	105.0
DREKF N=50 (*DREKF 50*)	0.03252	0.09978	46.2	0.03495	0.11363	106.6
DREKF N=10 beacon (*DREKF-D*)	0.01739	0.07056	44.8	0.01577	0.06306	104.9

## Data Availability

Not applicable.

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
