# Peer review of "Dual-Rate Extended Kalman Filter Based Path-Following Motion Control for an Unmanned Ground Vehicle: Realistic Simulation"

_sensors, 2021, doi:10.3390/s21227557_

Round 1
Reviewer 1 Report
This paper focus on the problem of the different sampling rates in the UGV sensor system, and proposes the dual-rate EKF to solve it. Also, a complete simulation system has been built to test the proposed method. The topic of this paper is interesting, and the content was well-organized. In my view, this paper deserves the high recommendation to publish on the journal.
Author Response
Reply: Thank you very much for your time and your positive evaluation.
Reviewer 2 Report
The paper is interesting and relevant. I would just make a few comments:
1- It is preferable to always highlight the results in their own section, instead of placing them in subsections, as is done (section 5.3).
2- “Remark 1” is used, with the text in italics, but there are no other remarks. As it is cited, later on, I believe it would be preferable to place the content, without the use of italics and without the word “remark”, as a subsection (in this case 3.3).
3- Table titles should be at the top and all table columns should have an explanatory heading.
4- In figure captions with items (a) and (b) write in such a way that the particular content is placed after the letter (eg (a) xxx... and (b) yyy...).
5- Expand PI and CAD acronyms the first time they are used.
Author Response
Reviewer 2. Comments and Suggestions for Authors
The paper is interesting and relevant.
Reply: Thank you very much for your careful review and your positive evaluation. We have coped with all the suggestions made by you. As a consequence, we think, the quality of the paper has been improved.
I would just make a few comments:
- It is preferable to always highlight the results in their own section, instead of placing them in subsections, as is done (section 5.3).
Reply: We agree with you. Now, we have included the results in section 6.
- “Remark 1” is used, with the text in italics, but there are no other remarks. As it is cited, later on, I believe it would be preferable to place the content, without the use of italics and without the word “remark”, as a subsection (in this case 3.3).
Reply: Right! Now, we have replaced Remark 1 with Section 3.3, and included a descriptive title about the subsection.
- Table titles should be at the top and all table columns should have an explanatory heading.
Reply: Done! Sorry for this format mistake.
- In figure captions with items (a) and (b) write in such a way that the particular content is placed after the letter (eg (a) xxx... and (b) yyy...).
Reply: Yes, this is another format mistake. Sorry about that. We have properly modified the captions.
- Expand PI and CAD acronyms the first time they are used.
Reply: Done! Thank you very much for your careful review.
Reviewer 3 Report
In this manuscript, an unmanned ground vehicle path-following motion control is proposed. Instead of using a dynamic controller, a dual-rate extended Kalman filter is used to estimate position and orientation measurement. In addition, the Simscape Multibody is applied to carry on simulation. However, there exist some problems in this manuscript.
- The title of each picture and its format should be checked and revised.
- Key parameters affecting the data should be analyzed and an underlying discussion of the results should be added to conclusions.
- Comparison experimental results of the existing methods and the proposed method are necessary for verifying the innovation of the proposed method.
I suggest to revise the paper to make it proper representative of the presented work.
Author Response
Reviewer 3. Comments and Suggestions for Authors
In this manuscript, an unmanned ground vehicle path-following motion control is proposed. Instead of using a dynamic controller, a dual-rate extended Kalman filter is used to estimate position and orientation measurement. In addition, the Simscape Multibody is applied to carry on simulation.
Reply: Thank you very much for your careful reading of the paper and your valuable inputs.
However, there exist some problems in this manuscript.
- The title of each picture and its format should be checked and revised.
Reply: We agree with you. Now, the caption of every figure has been centered. This change in the manuscript has not been highlighted in blue because it results in a general format change. Nevertheless, the improvements included in some figure descriptions (caption text) have been highlighted in blue to more easily view the change.
- Key parameters affecting the data should be analyzed and an underlying discussion of the results should be added to conclusions.
Reply: The simulator developed in this work includes a significant number of parameters to be tuned. They can be classified as dc motor parameters, friction parameters with the ground, and control and estimation algorithm parameters. All of them have been adjusted by experimentation. The dc motor parameters are adjusted by classical methods. Whereas friction parameters are tuned by using model-based optimization techniques, the control and estimation parameters are set by taking the nominal, desired behavior as the reference case, which is the performance reached by the fast single-rate control. For this reason, the control and estimation parameters may be considered as the set of key parameters of the simulator. Especially relevant to determine the desired behavior for the UGV are the velocity constant reference and look ahead distance used in the Pure Pursuit path tracking algorithm, and the covariance matrices formulated in the Dual-Rate Extended Kalman Filter (DREKF).
A sentence summarizing the paragraph above has been added to the Conclusions section.
- Comparison experimental results of the existing methods and the proposed method are necessary for verifying the innovation of the proposed method.
Reply: Yes, you are right. No experimental results are included in this paper. This aspect is planned as a future work, since it goes beyond the purpose of the present manuscript. However, it is worthy to note that satisfactory simulation results have been provided by considering a realistic model, which has been validated from real experiments (see Figure 10).
Regarding the comparison with other methods, indeed, we would like to emphasize that the introduced technique (the DREKF) has been compared both with traditional single-rate control solutions at different periods and with odometry estimation methods. Thank you very much for your concern.
I suggest to revise the paper to make it proper representative of the presented work.
Reply: After facing your instructive suggestions, we believe, the paper has been considerably improved. Thank you very much for your time.
Reviewer 4 Report
The paper presents the application of the dual-rate extended Kalman filter (DREKF) to control a two-wheeled unmanned ground vehicle (UGV). It can be used to create a fast-rate controller even when sensors provide data with different periods. Although DREKF is a known algorithm in literature, its application to UGV is an interesting problem. The paper is carefully prepared. The proposed controller and DREKF are well described. Correct mathematical notations and formulas are applied. The authors elaborated a comprehensive simulation model for UGVs by using Simscape Multibody modelling tool. The response from the simulation model was compared with the one of the real system and the simulation model proved to be realistic. Simulations were executed with various controllers. The results show that the UGV was able to precisely follow the predefined path by applying the proposed controller. A link to a video is included in the paper which presents the simulations. I did not found any typos in the paper.
Author Response

(The authors gave the same response as above.)

Round 2
Reviewer 3 Report
In this manuscript, dual-rate extended Kalman filter based path-following motion control for an unmanned ground vehicle was introduced. A Simscape Multibody™ (Matlab®/Simulink) model 12 has been developed for a realistic simulation, considering the contact forces between the wheels 13 and the ground, not included in the kinematic and dynamic UGV representation. However, there are some issues.
- What’s the parameter of the UGV?
- What’s the difference between the realistic simulation results and the physical results?
- Is the complex algorithm including dual-rate extended Kalman filter suitable for the UGV with small energy?
Author Response
In this manuscript, dual-rate extended Kalman filter based path-following motion control for an unmanned ground vehicle was introduced. A Simscape Multibody™ (Matlab®/Simulink) model has been developed for a realistic simulation, considering the contact forces between the wheels and the ground, not included in the kinematic and dynamic UGV representation.
Reply: Thank you very much for your careful reading of the paper and your instructive suggestions.
However, there are some issues.
1. What’s the parameter of the UGV?
Reply: Since the kinematic and dynamic models considered for the UGV do not consider the static and dynamic coefficients of friction with the ground, they have been adjusted via the simulator developed and by following the so-called data-driven modelling. Table 2 shows the friction coefficients of the UGV against the ground, which were obtained by experimentation, using the floor of our laboratory.
2. What’s the difference between the realistic simulation results and the physical results?
Reply: As we reached a very accurate model for the UGV (as depicted in Figure 10), the difference between the simulation results provided by the model and experimental results may be very subtle. The main differences may appear as a consequence of not considering in the model some possible irregularities of the floor or some possible imperfections of the measurement system. In order to try to consider these possible disturbances in the simulations, we additionally included measurement and process noises.
3. Is the complex algorithm including dual-rate extended Kalman filter suitable for the UGV with small energy?
Reply: Yes, it is feasible. As a future work, we are planning the real implementation of the control approach in a test-bed platform. By using FPGA or MCU based solutions, on-board Kalman filters can be usually implemented.